# Reducing greenhouse gas emissions of Amazon hydropower with strategic dam planning

Rafael M. Almeida [1], Qinru Shi[2], Jonathan M. Gomes-Selman[3], Xiaojian Wu [2,12], Yexiang Xue[2,13], Hector Angarita [4], Nathan Barros[5], Bruce R. Forsberg[6], Roosevelt García-Villacorta[1], Stephen K. Hamilton[7,8], John M. Melack[9], Mariana Montoya[10], Guillaume Perez[2], Suresh A. Sethi[11], Carla P. Gomes[2] & Alexander S. Flecker[1]

Hundreds of dams have been proposed throughout the Amazon basin, one of the world's largest untapped hydropower frontiers. While hydropower is a potentially clean source of renewable energy, some projects produce high greenhouse gas (GHG) emissions per unit electricity generated (carbon intensity). Here we show how carbon intensities of proposed Amazon upland dams (median = 39 kg $CO_2$eq $MWh^{-1}$, 100-year horizon) are often comparable with solar and wind energy, whereas some lowland dams (median = 133 kg $CO_2$eq $MWh^{-1}$) may exceed carbon intensities of fossil-fuel power plants. Based on 158 existing and 351 proposed dams, we present a multi-objective optimization framework showing that low-carbon expansion of Amazon hydropower relies on strategic planning, which is generally linked to placing dams in higher elevations and smaller streams. Ultimately, basin-scale dam planning that considers GHG emissions along with social and ecological externalities will be decisive for sustainable energy development where new hydropower is contemplated.

[1] Department of Ecology and Evolutionary Biology, Cornell University, Ithaca, NY 14853, USA. [2] Cornell University, Institute for Computational Sustainability, Ithaca, NY 14853, USA. [3] Department of Computer Science, Stanford University, Palo Alto, CA 94305, USA. [4] Stockholm Environment Institute Latin America, Bogota 110231, Colombia. [5] Department of Biology, Federal University of Juiz de Fora, Juiz de Fora 36036-900, Brazil. [6] National Institute of Amazonian Research (INPA), Manaus 69060-001, Brazil. [7] W.K. Kellogg Biological Station and Department of Integrative Biology, Michigan State University, Hickory Corners, MI 49060, USA. [8] Cary Institute of Ecosystem Studies, Millbrook, NY 12545, USA. [9] Bren School of Environmental Science and Management, University of California at Santa Barbara, Santa Barbara, CA 93106, USA. [10] Wildlife Conservation Society Peru, Lima 15048, Peru. [11] USGS New York Cooperative Fish and Wildlife Research Unit, Department of Natural Resources, Cornell University, Ithaca, NY 14853, USA. [12] Present address: Microsoft AI & Research, Sunnyvale, CA, USA. [13] Present address: Department of Computer Science, Purdue University, West Lafayette, IN, USA. Correspondence and requests for materials should be addressed to R.M.A. (email: rafaelmalmeida2@gmail.com) or to C.P.G. (email: gomes@cs.cornell.edu) or to A.S.F. (email: asf3@cornell.edu)

Hydropower has been promoted as a climate-friendly alternative to meet the world's growing electricity demand[1]. Globally, hydropower dam construction is expected to reach unprecedented rates in the coming decades, especially in countries with emerging economies[2]. One hotspot for future hydropower expansion is the Amazon[3–5], the world's largest river basin. Although dams have already been built in several regions of the basin, the Amazon hydropower potential remains largely untapped, and electricity generation is the primary motivation for new dam construction[2]. Existing evidence suggests that most global hydropower projects have total greenhouse gas (GHG) emissions per unit electricity generated (also known as carbon intensity, Table 1) within the range of other renewable energy sources like solar and wind power[6–8]. However, about 10% of the world's hydropower facilities emit as much GHGs per unit energy as conventional fossil-fueled power plants[6]. Some existing dams in the lowland Amazon have been shown to be up to ten times more carbon-intensive than coal-fired power plants[9–11]. In light of the expected boom in construction of new hydropower dams in the Amazon basin, it is critical to identify whether future dams will produce low-carbon energy.

GHG emissions from reservoirs stem primarily from the decomposition of organic matter that is either flooded, transferred to the reservoir via runoff and river input, or produced within the reservoir as aquatic plant and algal biomass[12]. Although part of the emissions would occur under natural pre-impoundment conditions, reservoirs generally result in net increases of both carbon dioxide ($CO_2$) and methane ($CH_4$) emissions to the atmosphere, and should thus be considered anthropogenic GHG sources[13,14]. $CH_4$ is the most important GHG produced in reservoirs and originates from bacterial decomposition of organic matter in anoxic water and sediment environments created by impoundment[13]. GHG emissions (Table 1) from reservoirs vary substantially over space and time[15,16], being positively correlated with temperature[17,18] and aquatic primary production[12], and negatively correlated with reservoir age[17,19]. Since total GHG emission is proportional to flooded area, the electricity generation capacity (installed capacity) per unit of reservoir flooded area, or power density (Table 1), is a key determinant of carbon intensity[8,9,20,21]. Hence, projects with low GHG emission (e.g., oligotrophic reservoirs[12]) can still have high carbon intensities if they produce low amounts of electricity per unit flooded area (i.e., low power density).

Environmental impact studies for new dams rarely consider GHG emissions, especially in developing countries where hydropower is currently expanding[22]. The problem is compounded by the piecemeal nature of these studies where each project is evaluated independently without considering the integrated effect of all existing and planned dams on basin-wide emissions. Here, we use a database of GHG fluxes for existing tropical and subtropical reservoirs[12] to calculate the range of carbon intensities expected for 351 proposed and 158 existing Amazon hydropower dams. To incorporate the time-related radiative forcing effect of $CH_4$, a potent GHG with an approximate atmospheric residence time of only about a decade, we conducted analyses of carbon intensities considering 20-year and 100-year time horizons. We found that carbon intensities vary by over two orders of magnitude from the lowest to the highest emitting dam, with projects in lower elevations and larger rivers being associated with higher emissions per unit electricity generated. Using a basin-wide optimization approach, we show that strategic dam planning could minimize aggregate carbon intensity as hydropower generation expands. Our approach can be adapted to different scales and could help Amazonian countries achieve their energy goals more sustainably.

## Results and Discussion

**Carbon intensities of proposed dams**. We estimate that existing Amazon hydropower reservoirs collectively emit 14 Tg $CO_2$eq per year over a 100-year time horizon (95% confidence interval (CI): 10–19), or ≈2% of the current total annual GHG emission from reservoirs globally[12]; if all 351 proposed dams are built, annual emissions from Amazon reservoirs would increase approximately fivefold (Supplementary Table 1). The carbon intensities of reservoirs that would be created by proposed dams differ markedly depending on whether dams are built in upland (> 500 m a.s.l.) or lowland reaches (Fig. 1).

Based on projections of the sustainable development scenario of the International Energy Agency's (IEA) World Energy Outlook 2017[23], we consider 80 kg $CO_2$eq MWh$^{-1}$ as a reference carbon intensity for sustainable electricity generation. This value is consistent with achieving the energy-related goals of the United Nations 2030 Agenda for Sustainable Development (2030 Agenda), which would reduce the collective carbon intensity of the global electricity sector from the current ≈500 kg $CO_2$eq MWh$^{-1}$ to ≈80 kg $CO_2$eq MWh$^{-1}$ in 2040. Our analysis indicates that most proposed upland dams (92% for a 100-year time horizon and 60% for a 20-year time horizon) would likely result in carbon intensities below 80 kg $CO_2$eq MWh$^{-1}$ (Fig. 1b, c). By contrast, only a minority of lowland dams would be expected to emit less than 80 kg $CO_2$eq MWh$^{-1}$ (36% for a 100-year time horizon and 14% for a 20-year time horizon). In fact, over a 20-year time horizon about 25% of the proposed lowland dams would likely be more carbon-intensive than coal-fired power plants (Fig. 1b).

Lowland dams have significantly higher carbon intensities due to their typically larger reservoir areas and innately lower power densities, whereas the steeper topography of high-elevation areas

---

**Table 1 Metrics commonly used to evaluate GHG emissions in hydropower projects**

| Metric | Units | Description |
|---|---|---|
| GHG flux | kg $CO_2$eq km$^{-2}$ d$^{-1}$ | The exchange of GHG, in $CO_2$ equivalents, at the reservoir air-water interface per unit of surface area over a certain time period. The direction of GHG flux can be from water to atmosphere (emission or efflux; positive value) or from atmosphere to water (uptake or influx; negative value). |
| Total GHG flux | Tg $CO_2$eq | GHG flux over a reference time period multiplied by the total reservoir area. The reference times considered here are a day and 1, 20, and 100 years (1 Tg = $10^{12}$ g). |
| Power density | MW km$^{-2}$ | The ratio of electricity generation capacity to reservoir flooded area. This metric reflects the strong link between GHG emissions and flooded area and is often used as a simple proxy for carbon intensity. |
| Carbon intensity | kg $CO_2$eq MWh$^{-1}$ | Also known as emission intensity or emission factor. $CO_2$-equivalent emissions produced per unit electricity generated. This metric is used to compare emissions performance across projects of different sizes, and also among electricity sources. |

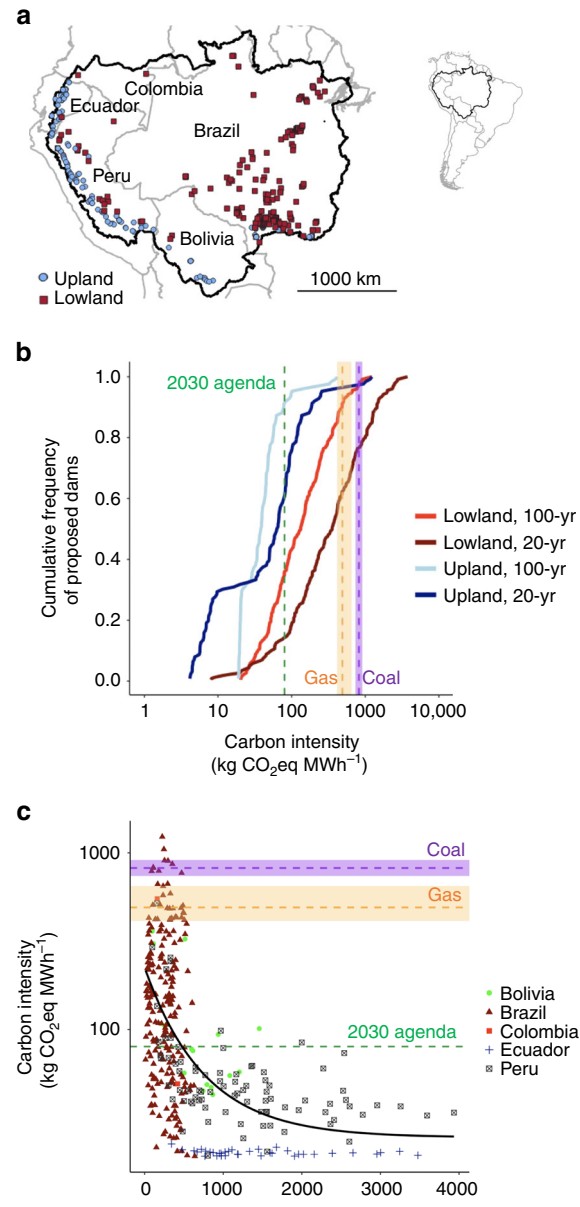

**Fig. 1** Carbon intensity of proposed Amazon hydropower dams. A total of 351 dams (>1 MW) have been proposed in five different countries of the Amazon basin, 65% in elevations below 500 m a.s.l. (lowland) and 35% above 500 m a.s.l. (upland) (**a**). **b** Cumulative frequency of lowland (< 500 m a.s.l.) and upland (> 500 m a.s.l.) dams proposed for the Amazon basin with respect to predicted carbon intensities over 20- and 100-year time horizons. **c** Carbon intensities of proposed dams (100-year time horizon) plotted against elevation above sea level. Point colors correspond to the countries where each dam is located. Green dashed lines (80 kg $CO_2$eq $MWh^{-1}$) indicate the projected carbon intensity of the global electricity sector in 2040 based on a scenario consistent with the UN 2030 Agenda for Sustainable Development[23]. The ranges of carbon intensities of coal- and natural gas-fired power plants reported by the IPCC are shown in the purple and orange areas, respectively, with medians indicated by horizontal dashed lines[7]

favors hydropower projects with higher power densities. This explains why the largest number of Amazon dams with high carbon intensities occurs in Brazil, a predominantly lowland country, whereas dams with lower carbon intensities are concentrated in mountainous parts of Bolivia, Ecuador, and Peru

(Fig. 1c). Notably, while it has recently been suggested that dams can mitigate natural GHG emissions from downstream floodplain wetlands by reducing the extent and duration of inundation[1], hydropower dams that can regulate inundation of downstream wetlands are typically those in lowland reaches, which generally implies lower power density and hence high carbon intensities for such projects. In addition, it is critical to understand whether lowland dams are more likely to create reservoirs enriched in nutrients such as phosphorus and nitrogen, which would increase aquatic primary production and consequently GHG emissions[12,24], thereby increasing their carbon intensities.

**Achieving low-carbon hydropower with strategic planning.** Our findings suggest that Amazon hydropower must be developed strategically on a basin-wide scale to achieve low-carbon energy goals. We therefore performed a multi-objective optimization to determine the Pareto-optimal frontier[25], which defines the set of solutions (i.e., dam portfolios) that minimizes total basin-wide GHG emissions while satisfying varying hydroelectricity generation goals (Supplementary Fig. 1). Our computational framework adapts and parallelizes previously proposed algorithms[26,27] to compute the exact (provably optimal) Pareto frontier for $2^{351}$ ($\approx10^{105}$) possible combinations of proposed Amazon dams in very fast computational time (< 10 min) (Supplementary Fig. 2).

Our multi-objective optimization indicates that if future hydropower dams are selected optimally, it will be possible to develop ≈80% (75 GW) of the total proposed electricity generation capacity while creating a portfolio of new dams with an aggregate carbon intensity below 80 kg $CO_2$eq $MWh^{-1}$ over a 100-year time horizon (Fig. 2a, b). Conversely, uncoordinated planning may result in portfolios of new dams with collective carbon intensities incompatible with sustainable energy goals (Fig. 2a, b). For instance, suboptimally exploiting about 15 GW of the total proposed installed capacity—which is equivalent to the current installed capacity of the entire electricity sector of Bolivia, Ecuador and Peru—could result in hydropower portfolios as carbon-intensive as equivalent electricity generation by fossil-fuel sources (Fig. 2a, b, e). Optimal planning, however, would allow the exploitation of 15 GW through a portfolio of new dams emitting < 25 kg $CO_2$eq $MWh^{-1}$ for a 100-year time horizon, which is below the carbon intensity of a typical solar power plant[7] (Fig. 2a, b, f). Thus, the ability of hydropower to mitigate climate change[1] relies critically on strategic dam portfolio planning so as to avoid carbon-intensive projects, especially over short time horizons (Fig. 2a).

Building dams without basin-wide coordination has led to a current Amazon dam portfolio with a collective carbon intensity of ≈200 kg $CO_2$eq $MWh^{-1}$ (20-year time horizon) and ≈90 kg $CO_2$eq $MWh^{-1}$ (100-year time horizon) (Fig. 2c, d). Optimal selection of future dams can lead to significant improvements, lowering the overall carbon intensity of Amazon hydropower (Fig. 2c, d). After ≈75 GW of the proposed Amazon hydropower potential is tapped, however, it will not be possible to add extra dams without increasing the corresponding carbon intensity of the portfolio (Fig. 2c, d). This would occur because all of the most efficient proposed projects would have been selected; thus, tapping more energy thereafter implies selecting more dams on higher-order streams at lower elevations, which tend to have higher carbon intensities (Fig. 3).

The need for strategic planning to balance energy and water management benefits provided by dams with associated social and environmental externalities is becoming increasingly apparent[25,28–34]. For instance, a study in a large tributary basin to the Mekong River, the largest river in Southeast Asia, has

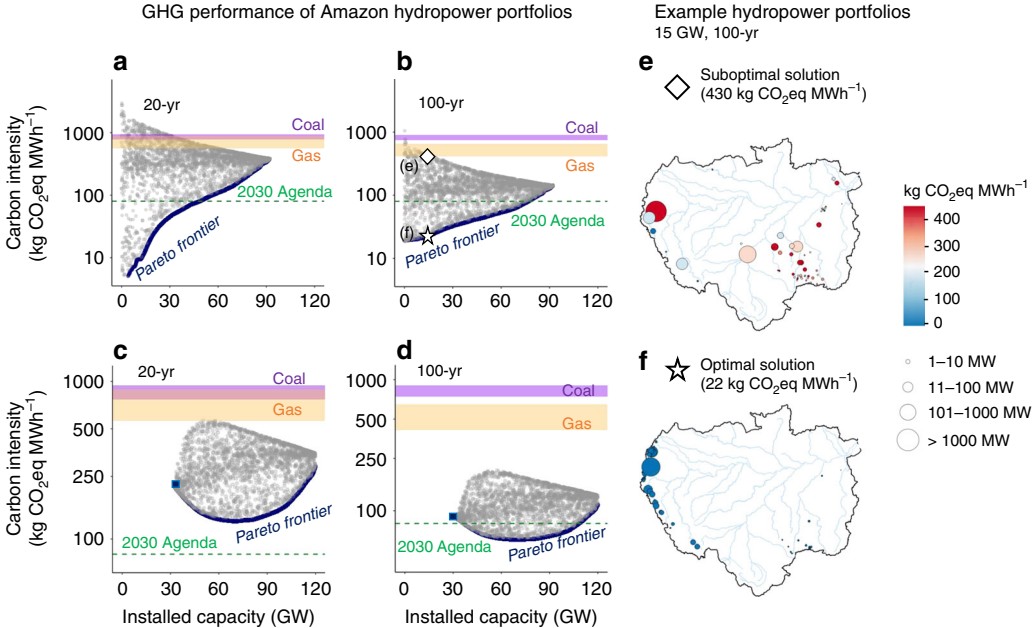

**Fig. 2** Tradeoffs between hydropower generation and carbon intensity for portfolios of proposed Amazon dams. In **a–d**, each point represents a portfolio of dams. The optimal dam portfolios for each value of installed capacity (Pareto frontier) are shown in dark blue, and randomly generated suboptimal dam portfolios are shown by gray symbols. **a** and **b** show carbon intensity outcomes considering only the 351 proposed dams over 20-year and 100-year time horizons, whereas **c** and **d** show outcomes starting from the existing set of 158 Amazon dams (blue square; current installed capacity = 33 GW for an emission of 213 kg $CO_2$eq $MWh^{-1}$ over a 20-year horizon and 87 kg $CO_2$eq $MWh^{-1}$ over a 100-year horizon). The ranges of carbon intensities of coal- and natural gas-fired power plants reported by the IPCC are shown in the purple and orange bands, respectively[7]. The carbon intensity of electricity produced from natural gas is closer to that of coal over shorter time frames due to its higher methane emissions. The green dashed line indicates the projected carbon intensity of the global electricity sector based on a scenario consistent with the UN 2030 Agenda for Sustainable Development[23]. A suboptimal and a Pareto-optimal dam portfolio, both with the same installed capacity (15 GW) but with contrasting carbon intensities (100-year time horizon), are illustrated in (**e**) and (**f**)

demonstrated that strategic planning would have allowed the exploitation of 70% of the basin's hydropower potential while trapping only 20% of the river's sand-sized (64 μm to 2 mm) sediment load, which is critical for downstream geomorphology, floodplain development and aquatic biota. However, project-by-project hydropower development implemented in the region has led to trapping of more than 90% of the sand load while exploiting only 50% of the hydropower potential[33]. Similar concepts of strategic planning have been applied to optimize dam removal strategies. For instance, a study in the Willamette River basin in the western US has shown that removing 12 existing dams would reconnect over 50% of the river network while sacrificing <2% of current hydropower and water-storage capacity[28].

**Climate-friendly hydropower projects.** Because the carbon intensity of hydroelectric dams is strongly linked to power density[9,20,21], power density is the criterion employed by the Clean Development Mechanism of the UN Framework Convention on Climate Change to finance and grant carbon credits to hydropower projects[35]. Projects with power densities above 4 MW $km^{-2}$ are eligible for credits and GHG emissions from candidate projects with power densities above 10 MW $km^{-2}$ are assumed to be negligible over 100-year horizons. While power density may provide a convenient sustainable energy metric, natural variability in GHG emissions observed in reservoirs can lead to differences in carbon intensities for dams with comparable power densities. We plotted power densities against our predicted carbon intensities to examine what densities may satisfy sustainable energy goals (i.e., < 80 kg $CO_2$eq $MWh^{-1}$). For

a 100-year time horizon, power densities above 6.7 MW $km^{-2}$ (95% CI: 4.5–9.5) were associated with projects emitting <80 kg $CO_2$eq $MWh^{-1}$ (Fig. 4a). The lower bound of the 95% CI (4.5 MW $km^{-2}$) suggests that the Clean Development Mechanism lending criterion of 4 MW $km^{-2}$ avoids most carbon-intensive projects. The more conservative upper bound of the 95% CI indicates that projects are very likely to emit <80 kg $CO_2$eq $MWh^{-1}$ only when power densities exceed 9.5 MW $km^{-2}$; about half of the proposed Amazon dams have power densities below 9.5 MW $km^{-2}$ (Fig. 4a). Considering a 20-year time horizon for carbon intensities causes approximately a threefold increase in the power density threshold for designating climate-friendly projects (Fig. 4b).

On a basin scale, prioritizing projects with high power densities can attenuate carbon intensities of future hydropower dam portfolios. However, mitigation measures can also reduce the carbon intensities of individual projects. Tackling internal and external sources of organic matter supporting $CH_4$ production in reservoirs is key. Previous studies suggest that reducing nutrient inputs to reservoirs[12,36] and clearing terrestrial vegetation prior to flooding[20] can significantly decrease carbon intensities of hydropower projects. In addition, project-scale improvements to power densities can make future hydropower projects less carbon-intensive[20], including alternative project designs that sacrifice a fraction of power generation to favor disproportionately smaller reservoir flooded areas, which would increase power density and hence reduce carbon intensity. Finally, retrofitting existing hydropower turbines with more efficient designs can increase electricity generation by up to 30% without requiring additional flooded area[31], thus contributing to lower carbon intensities in the hydropower sector.

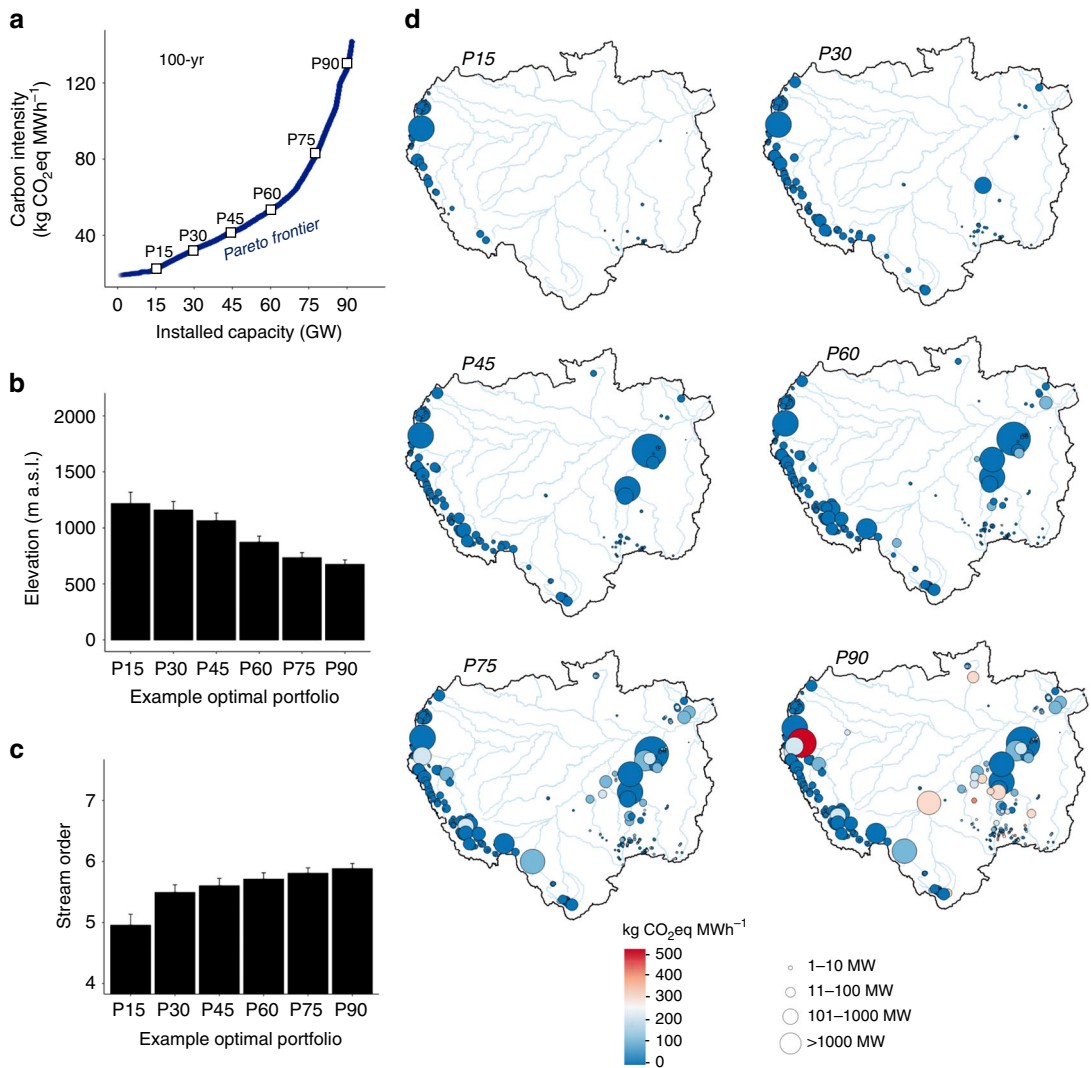

**Fig. 3** Characterization of optimal dam configurations as electricity generation increases. **a** Carbon intensity outcomes (100-year time horizon) of optimal dam portfolios for different values of installed capacity considering the 351 proposed Amazon dams; squares indicate six example reference portfolios spanning increasing installed capacity (P) from 15–90 GW. The mean (±s.e.m.) elevation (**b**) of dams decreases, and stream order at dam locations (**c**) increases, as optimal portfolios target greater total installed capacity and subsequently include more dams in lowland areas of the Amazon basin (**d**). Stream order is a metric used in hydrology to indicate the level of branching in a river network, where increasing stream order correlates with increasing channel size and discharge

**Moving forward**. Our findings point to the complexities of utilizing hydropower as an energy source compatible with climate change mitigation. Integrated regional assessments of GHG emissions can help identify portfolios of dams that are consistent with low-carbon energy goals. Although our study has focused on Amazon dams, our approach can be adapted to other regions where hydropower is rapidly expanding, including the Balkans and major river basins such as the Congo, the Mekong, the Ganges-Brahmaputra, and the Yangtze[2].

Carbon intensity is a key criterion for sustainable energy planning. However, we emphasize that hydropower dams have a wide range of additional interactions with social and ecological systems, and some dams may have other purposes such as water supply, flood control, and recreation. Dam construction can lead to social disruptions[37] and seriously compromise a variety of ecosystem services and processes[38] including altered natural flow and flood regimes[39,40], reduced sediment[33] and nutrient[41] supply to downstream waters, blockage of fish migrations[3], deterioration of habitat connectivity[42,43], and loss of biodiversity[42–44]. Ultimately, a broader suite of criteria including consideration of

alternative energy sources will be needed to fully integrate the social and ecological externalities into strategic hydropower planning, ideally using a multicriteria optimization framework building on the approach we employed in this study.

## Methods

**Amazon dams database**. Geographic location, elevation and technical data including installed capacity and flooded area for proposed and existing dams were obtained from published databases on existing and proposed Amazon dams[3,45]. Our database incorporated information from recent national government databases for countries where updated inventory data were readily available[46,47]. We calculated the level of branching in the river network using the Strahler stream order method[48].

There are 158 existing dams, either operating or under construction, with over 1 MW of installed capacity in the Amazon basin, totaling 32,608 MW of electricity generation capacity with an average of 206 MW per dam (range: 1–11,233 MW). We identified 351 proposed dams in various stages of inventory, planning and licensing (installed capacity >1 MW). The proposed dams have a combined electricity generation capacity of 91,887 MW, on average 262 MW per dam (range: 1–6133 MW). Watershed areas above each dam were estimated from a digital elevation model of the region. Existing and proposed dams were categorized as upland or lowland using a cutoff of 500 m a.s.l.[49]. In some cases (26% of dams), information on flooded areas was unavailable. For existing reservoirs without

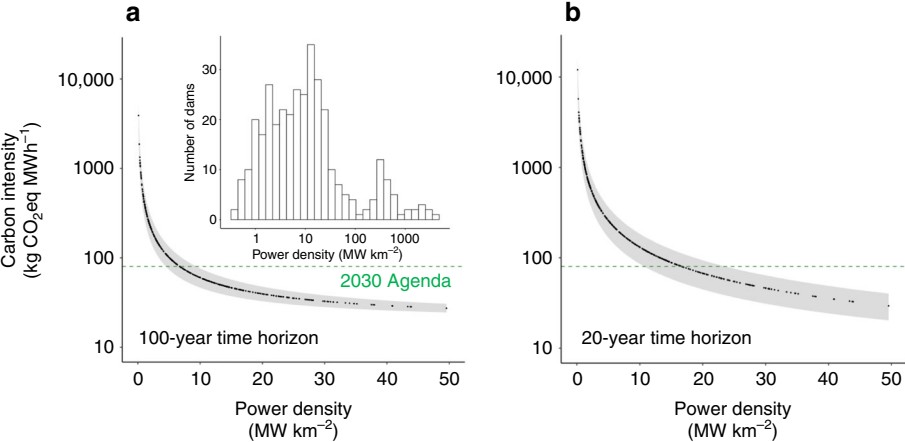

**Fig. 4** Low-carbon power densities for Amazon hydropower. Power density is a key determinant of carbon intensity. We plotted the functional relationship between power density and carbon intensity for existing and proposed Amazon dams over a **a** 100-year and **b** 20-year time horizon. Shaded areas reflect uncertainty about GHG fluxes (95% bootstrap confidence region, see Methods). Points below the green line indicate projects with carbon intensities that satisfy the reference carbon intensity for sustainable electricity production (80 kg $CO_2$eq $MWh^{-1}$). To improve visualization, we omitted projects with power densities above 50 MW $km^{-2}$ (≈25% of the dams). The inset figure in (**a**) shows the frequency distribution of the power densities of all proposed Amazon dams

reported flooded areas, we quantified flooded areas from satellite imagery (Google Earth Pro 7.3.2.5776). For proposed dams with missing information, we used available flooded areas as a training dataset to develop a multiple regression model including country, watershed area, installed capacity, and elevation as covariates to estimate flooded areas (Supplementary Fig. 3a). The predictive power of the regression model was high; however, we also ran sensitivity analyses to confirm that our main conclusions were robust to the inclusion of estimated flooded areas for the subset of dams with missing data (Supplementary Fig. 3b).

**Time horizon of the analyses.** To compare the radiative forcing effects of GHGs with different warming potentials and atmospheric residence times, an index termed Global Warming Potential (GWP) is typically used. The GWP measures the relative amount of energy the emission of a gas will absorb over a given period of time in relation to the same amount of $CO_2$. The most widely used time horizon for GWP of atmospheric gases is 100 years, but shorter time frames are particularly appropriate for interpreting the climate effects of certain activities when short-lived gases are to be prioritized. This is the case of $CH_4$, which remains in the atmosphere for approximately a decade but has a large radiative forcing effect. Therefore, all of our analyses also consider a 20-year time horizon in addition to the commonly used 100-year time horizon. In terms of radiative forcing, $CH_4$ is the predominant GHG emitted from hydropower reservoirs, and the general temporal pattern of GHG emissions from dams indicates that emissions peak in the first decade after damming and then fall to lower levels that remain somewhat constant over time[13]. Therefore, the high potential of dams to cause warming over short timescales gets underrepresented when the GHG footprint of dams is assessed only over long time horizons. We converted $CH_4$ emissions to $CO_2$-equivalents using a GWP of 34 over 100 years and 86 over 20 years[50].

**Carbon intensity estimates.** The carbon intensity (also referred to as emission intensity or emission factor) of power sources measures the net GHG emission per unit electricity generated (kg $CO_2$eq $MWh^{-1}$). We combined project-specific data on flooded areas and installed capacity from our Amazon dams database with 48 $CO_2$ and 38 $CH_4$ published flux estimates for tropical and subtropical reservoirs[12] to calculate carbon intensity ranges for all existing and proposed Amazon dams. To calculate the carbon intensity of a given dam, we first calculated total GHG flux as follows:

$$\text{TE}_{\text{dam}} = A_{\text{dam}} \times \left( \text{net}_{CO_2} \times F_{CO_2,\text{dam}} + \text{net}_{CH_4} \times F_{CH_4,\text{dam}} \times \text{GWP}_{CH_4} \right) \times \left( 1 + R_{\text{downstream}} \right) \quad (1)$$

where $\text{TE}_{\text{dam}}$ is the total GHG flux (kg $CO_2$eq $d^{-1}$), with positive values denoting emission (water-to-atmosphere flux) and negative values denoting uptake (atmosphere-to-water flux); $A_{\text{dam}}$ is the reservoir flooded area ($km^2$); $F_{CO_2,\text{dam}}$ is the $CO_2$ flux (kg $CO_2$ $km^{-2}$ $d^{-1}$); $F_{CH_4,\text{dam}}$ is the $CH_4$ flux (kg $CH_4$ $km^{-2}$ $d^{-1}$); $\text{GWP}_{CH_4}$ is a conversion factor for the global warming potential of $CH_4$ over the corresponding time horizon (20 or 100 years) to transform kg $CH_4$ $km^{-2}$ $d^{-1}$ to kg $CO_2$eq $km^{-2}$ $d^{-1}$; $R_{\text{downstream}}$ is a constant representing the ratio of downstream emissions to reservoir-surface emissions, estimated to be 17%[51]. We multiplied $CO_2$ fluxes by a discount factor of 0.25 ($\text{net}_{CO_2}$) and $CH_4$ fluxes by 0.90 ($\text{net}_{CH_4}$) to account only for the net (anthropogenic) change in GHG emissions associated with reservoir creation (see details below). We

then calculated total electricity generation as follows:

$$\text{EG}_{\text{dam}} = \text{Cap}_{\text{dam}} \times 24 \times P_{\text{Cap}} \quad (2)$$

where $\text{EG}_{\text{dam}}$ is the total electricity generation of a given dam over a day (MWh $d^{-1}$); $\text{Cap}_{\text{dam}}$ is the installed capacity (MW), which was multiplied by 24 to obtain the energy output in 24 h and to have numerator and denominator units of Eq. (3) in the same time unit; and $P_{\text{Cap}}$ is a constant representing the capacity factor (0.5727), which denotes the effective electricity generation as a proportion of installed capacity, and was derived from an empirical relationship between data in our database on existing Amazon dams. Carbon intensity ($\text{CI}_{\text{dam}}$, kg $CO_2$eq $MWh^{-1}$) is then calculated as:

$$\text{CI}_{\text{dam}} = \frac{\text{TE}_{\text{dam}}}{\text{EG}_{\text{dam}}} + \text{CI}_{\text{construction}} \quad (3)$$

where $\text{CI}_{\text{construction}}$ is a constant representing the carbon intensity associated with construction and infrastructure of hydropower dams (19 kg $CO_2$eq $MWh^{-1}$ for a 100-year time horizon)[7].

Uncertainty in estimated carbon intensities for proposed Amazon dams is largely influenced by variability in the GHG flux input data (i.e., $F_{CO_2,\text{dam}}$ and $F_{CH_4,\text{dam}}$ in Eq. (1)). Thus, for each Amazon dam, we generated 10,000 carbon intensity predictions through the implementation of a bootstrapping procedure that randomly resampled with equal probability from the dataset of published $CO_2$ and $CH_4$ fluxes from tropical and subtropical reservoirs[12]. $CH_4$ fluxes from these reservoirs included both ebullition (bubbles rising directly from sediments) and diffusion. Our bootstrapped ranges of carbon intensities therefore reflect project-to-project variability in GHG flux rates as observed for existing tropical and subtropical dams. The $CO_2$ and $CH_4$ fluxes measured for single dams[12] were found to be uncorrelated ($r = 0.19$, $p = 0.16$), which allowed us to combine independently resampled $CO_2$ and $CH_4$ fluxes. Variation in calculated carbon intensity among dams is essentially driven by two parameters: installed capacity and flooded areas. Supplementary Fig. 4 shows examples of the bootstrapping output for two existing dams with contrasting power densities. Emissions results presented in the main text are based on mean and 95% confidence intervals for bootstrapped values.

Our calculations incorporate the net change in GHG fluxes resulting from the transformation of a riverine landscape into a reservoir by dam construction. The most comprehensive review on GHG emissions from reservoirs, which we used to support our analysis, reported gross fluxes[12]. To assess the net change in GHG fluxes resulting from the creation of a reservoir, emissions that would have existed under pre-impoundment conditions have to be discounted from the gross fluxes. Although conceptually simple, disentangling natural and anthropogenic reservoir emissions is a complex task with limited empirical support[13]. A recent review suggested that it is reasonable to assume that practically all $CH_4$ emissions from global reservoirs are new and therefore anthropogenic, whereas the majority of $CO_2$ emissions (perhaps ≈75%) over a 100-year time horizon would take place even without the reservoir creation[13]. In our analysis, we conservatively assumed that 75% of reservoir $CO_2$ emissions and 10% of $CH_4$ emissions reflect natural pre-impoundment emissions, and thus we incorporated these corrections in Eq. (1) ($\text{net}_{CO_2}$ and $\text{net}_{CH_4}$). For a particular reservoir, the percentage of $CH_4$ emissions that can be attributed to reservoir creation depends in part on the preexisting environments that become inundated; floodplains and other wetlands would have higher $CH_4$ emissions rates than non-wetland environments[52,53]. We use the 10%

estimate in our analysis because preexisting land cover information for all of the existing and proposed reservoirs in the Amazon is not available. We ran sensitivity analyses to verify how much these assumptions affect our results (Supplementary Fig. 5). Emissions of nitrous oxide ($N_2O$) can also occur in reservoirs; however, this gas was not considered in our analysis because $N_2O$ emissions generally represent < 5% of the total gross $CO_2$-equivalent emissions from impoundments[12], and because Amazon soils have naturally high rates of $N_2O$ emission, such that net increases in $N_2O$ emissions associated with dams are expected to be relatively low[54].

Previous studies indicate that reservoir GHG emissions vary as a function of temperature[18] and therefore latitude[17], with low-latitude dams generally emitting more GHG per unit area. Thus, we used flux information only from tropical and subtropical dams in the global reservoir emissions database to represent the latitudinal range of dam projects proposed in the Amazon[12]. Sensitivity analyses indicated that carbon intensities would not change substantially if fluxes from tropical dams only or dams from all climates (with most dams being located in northern temperate zones) were utilized instead of the subset that we adopted (Supplementary Fig. 6).

The increased rate of GHG emissions varies over the lifetime of a hydropower dam, with peak fluxes occurring in the first years after damming due to the decomposition of flooded biomass, followed by a protracted period of lower fluxes due to decomposition of soil organic matter, continuing river inputs, and new aquatic primary production[9,14,19]. The reported GHG flux measurements for tropical and subtropical dams[12] refer to dams on average > 30 years old, which means that they reflect GHG fluxes that miss the large pulse of emissions anticipated when dam reservoirs are first flooded. To account for the initial pulse of emissions from a hydropower project, we applied multiplier factors to the reported emissions associated with the first 5 years post-damming (300% for years 1–3, 200% for years 4–5) for a given dam for which we predict a carbon intensity, based upon the emissions profile from an existing Amazon reservoir[19,20].

**Validation of estimated carbon intensities**. To assess the validity of our approach to generating predicted carbon intensities for Amazon dams, we compared our estimated carbon intensities against intensities calculated using reported measurements of $CO_2$ and $CH_4$ fluxes for operational Amazon dams in ref. [12] ($n$ = 6). Our predictions were in reasonable agreement with observed carbon intensities (Supplementary Fig. 7), which was supported by a paired $t$-test between observed and mean modeled values ($t = -1.0$, two-tailed $P = 0.34$, degrees of freedom = 5).

**Carbon intensity of electricity sources**. The International Energy Agency (IEA) releases an annual report on the status and trends of global energy (World Energy Outlook), which includes carbon intensities anticipated under a range of global energy development scenarios[23]. To place proposed hydropower dams in the Amazon in a global energy production context, we used benchmarks from the IEA 2040 Sustainable Development Scenario, which portrays a decarbonized global electricity sector to meet the United Nations 2030 Agenda for Sustainable Development goals[55]. The IEA report suggests that a decarbonized global electricity sector should emit about 80 kg $CO_2$eq $MWh^{-1}$ in 2040, which is representative of a power mix sustained by renewables such as solar and wind power, as well as low-carbon hydropower plants. We also directly compared our calculated carbon intensities for Amazon hydropower dams against those reported for alternative energy technologies by the Intergovernmental Panel on Climate Change (IPCC), including coal-fired, combined-cycle natural gas-fired, and solar power plants[7]. The carbon intensities reported by the IPCC are for a 100-year time horizon. Owing to $CH_4$ emissions, carbon intensities of natural gas and coal are at least 37 and 4% higher over a 20-year time horizon, respectively, compared with a 100-year time horizon[56]. We applied 37 and 4% correction factors to obtain carbon intensities for natural gas and coal over 20 years.

**Tradeoff analysis and computation of the Pareto frontier**. To analyze the tradeoffs between electricity generation capacity and GHG emissions, we computed the Pareto frontier with respect to two criteria. The Pareto frontier is a function that identifies for a given installed capacity target the portfolio (or combination) of dams with the lowest amount of GHG emissions, or conversely, for a given GHG emission target, the portfolio of dams with the highest installed capacity. In our case, considering the 351 proposed dams in the Amazon basin, the possible portfolios of dams are: the empty portfolio that builds none of the proposed dams, 351 singleton portfolios with only one dam, 61,425 portfolios with two dams each $\binom{351}{2}$, 7,145,775 portfolios with three dams each $\binom{351}{3}$, and so on, until we reach the final portfolio comprising all 351 dams.

The application of the Pareto frontier is illustrated in the following scenarios. In Scenario 1, portfolio A has an installed capacity of 20,000 MW and carbon intensity of 90 kg $CO_2$eq $MWh^{-1}$, whereas portfolio B has an installed capacity of 20,000 MW and carbon intensity of 100 kg $CO_2$eq $MWh^{-1}$; we say that portfolio A dominates portfolio B since portfolio A has a lower carbon intensity for the same electricity generation capacity. In Scenario 2, portfolio A has an installed capacity of 20,000 MW and carbon intensity of 90 kg $CO_2$eq $MWh^{-1}$, whereas portfolio B has an installed capacity of 18,000 MW and carbon intensity of 100 kg $CO_2$eq $MWh^{-1}$; in that case we say that portfolio A dominates portfolio B since portfolio A has lower carbon intensity and higher electricity generation capacity. In Scenario 3, portfolio A has an installed capacity of 20,000 MW and carbon intensity of 90 kg $CO_2$eq $MWh^{-1}$, whereas portfolio B has an installed capacity of 18,000 MW and carbon intensity of 85 kg $CO_2$eq $MWh^{-1}$; in this scenario neither portfolio dominates the other. The Pareto frontier is then defined as the set of all portfolios of dams that are not dominated by any other portfolio.

Computing the exact Pareto frontier is a challenging computational problem, referred to as non-deterministic polynomial-time hard (NP-hard) problem, which means that in the worst case the computational time increases exponentially as a function of the number of dams[27]. Our framework for computing the exact (i.e., provably optimal) and approximate (with optimality guarantees) Pareto frontier exploits the tree structure of river networks[26,27], extending previously proposed algorithms for single-objective optimization stochastic network design in bidirected trees[57,58] to multi-objective optimization and computation of the Pareto frontier. In this approach, the river network is converted into a more abstract tree structure, whereby a node corresponds to a continuous section of the river uninterrupted by existing or proposed hydropower dams and an edge represents a proposed or an existing dam. This abstract tree structure is used by our dynamic-programming algorithm for the sequence of the merging and pruning of Pareto-optimal solutions.

The dynamic-programming approach recursively computes the Pareto-optimal partial solutions from leaf nodes up to the root[26,27]. The key insight is that at a given node $u$, we only need to keep the Pareto non-dominated partial solutions and we can therefore eliminate suboptimal (dominated) solutions. To increase incremental pruning, we convert the original tree into an equivalent binary tree. Given a binary tree, we first compute non-dominated Pareto solutions for the two children of the given parent node $u$, enumerate the partial solutions from the children and consider the four possible different combinations of whether to include each of the dams associated with each edge from the children. We then compute the objective values for the different extended partial solutions and add them to the set of overall partial solutions. Finally, we remove all dominated partial solutions from this set, so that the remaining partial solutions are Pareto-optimal for the parent node. This procedure allows us to systematically explore the entire search space of possible Pareto-optimal solutions. To prevent memory overflow in response to the large number of partial Pareto solutions considered, the algorithm batches partial solutions at each node and is parallelized to speed up the approach. We do not assume spatial dependencies among reservoirs when optimizing hydropower for GHGs, but consideration of spatial dependence may be critical for other environmental criteria (e.g., fish migrations or sediment retention), and our algorithm has the ability to solve problems where spatial dependence is important to consider.

In addition to computing the exact Pareto frontier, our dynamic-programming approach can provide a fully polynomial-time approximation scheme (FPTAS) by applying a rounding technique to the exact algorithm. The FPTAS finds a polynomially succinct solution set, which approximates the Pareto frontier within an arbitrary small factor $\varepsilon$ and runs in time that is polynomial in the size of the instance and $1/\varepsilon$[26,27]. The exact algorithm guarantees to find all optimal portfolios on the Pareto frontier. The approximate algorithm finds fewer portfolios but guarantees that every portfolio on the exact Pareto frontier is $\varepsilon$-approximately dominated by one of the portfolios on the approximate Pareto frontier. The algorithm used in our framework adapts and parallelizes a dynamic-programming based algorithm for the exact and approximate Pareto frontier. More computational details concerning our approach can be found in ref. [26,27], and the code is publicly available (see Code Availability section).

Compared with previous approaches used to compute the Pareto frontiers for dam placement, our algorithm provides coverage optimality guarantees and runs faster. Importantly, we also show that the approximate version of our algorithm is guaranteed to run in polynomial time (Supplementary Fig. 2). The computation of the exact Pareto frontier for the 351 proposed dams takes 8.6 min (wall-clock time, 8 threads; ≈1 h CPU time) and produces 83,108 non-dominated portfolios. Computing the $\varepsilon$-approximate Pareto frontier with 99% accuracy (i.e., $\varepsilon = 0.01$) for the 351 proposed dams takes 1.5 min wall-clock time (8 threads, ≈7 min CPU time) and produces 66,312 non-dominated portfolios. Except for Supplementary Fig. 2, all results presented here are based on the exact Pareto frontier. Finally, we also generated random suboptimal portfolios to compare with the Pareto-optimal ones. Due to the large number of all possible portfolios ($≈10^{105}$), we show only a subset of the suboptimal portfolios.

## Data availability
All relevant data are publicly available in the supplementary materials and online data repositories, and are available from the authors.

## Code availability
The Pareto optimization code can be downloaded from Cornell University's Institute for Computational Sustainability website (http://www.cs.cornell.edu/gomes/udiscoverit/downloads/hydro-pareto-tree-dp-2c/gomes-selman-et-al-dp-amazon-e-ghg-naturecommunications-2019.zip).

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

## Acknowledgements

This work was supported by Cornell University's Atkinson Center through a Post-doctoral Fellowship in Sustainability to R.M.A. and the Atkinson Academic Venture Fund. The computational work was supported in part by an NSF Expeditions in Computing award (CCF-1522054) and a Future of Life Institute grant. The computations for the Pareto frontier were performed using the AI for Discovery Avatar (AIDA) computer

cluster funded by an Army Research Office (ARO), Defense University Research Instrumentation Program (DURIP) award (W911NF-17-1-0187). We are grateful to Bridget Deemer and Henriette Jager for providing valuable comments and suggestions. We thank Scott Steinschneider and all participants of the Amazon Dams Computational Sustainability Working Group, and appreciate helpful comments from Robert Howarth, Peter McIntyre, Fábio Roland, and the Cornell Limnology Group.

## Author contributions

R.M.A., A.S.F., C.P.G., B.R.F., and S.K.H. conceived this study. R.M.A., S.A.S., and N.B. ran the carbon intensity analysis. C.P.G., Q.S., J.M.G.-S, X.W., and Y.X. designed and developed the computational framework for the Pareto optimization for this work. J.M.G.-S., Q.S., and G.P. implemented the algorithms for the Pareto optimization. Q.S. ran all the computations and experimental work for the Pareto optimization. R.G.-V., R.M.A., Q.S., and A.S.F. compiled and curated the hydropower dam dataset. R.M.A., Q.S., A.S.F., C.P.G., R.G.-V., B.R.F., S.K.H., N.B., S.A.S., H.A., M.M., and J.M.M. worked on the interpretation of the data. C.P.G. and Q.S. wrote the methods for the Pareto optimization. R.M.A. wrote the paper in close collaboration with A.S.F., B.R.F., and C.P.G., and with substantive revision by all authors.

## Additional information

**Competing interests:** The authors declare no competing interests.

