## [Peer Review File · Nature Communications]

Reviewers' comments:

Reviewer #1 (Remarks to the Author):

Review of 'Reducing greenhouse gas emissions of Amazon hydropower with optimal dam planning' by Almeida et al.

The study seeks to limit the GHG emissions of alternative combinations of proposed and existing hydropower project in the Amazon basin to meet a specified target. GHG emissions are estimated for all existing and proposed dams. Estimates are validated and shown to be reasonably good. Reservoirs are shown to vary in GHG emission from less than emissions from wind energy to more than fossil energy sources.

A new multi-objective optimization technique is used to develop a Pareto-optimal frontier of solutions consisting of portfolios of hydropower dams and associated GHG emissions and power capacity. This new optimization method takes advantage of river network structure and has guarantees of producing exact solutions.

Major points

1. The authors should consider reminding readers that there are many other aspects of sustainability that should be considered when planning a configuration of hydropower projects in the Abstract. This is mentioned later, in the 'Moving forward' section. In a hot spot of biodiversity like the Amazon basin, short-term methane emissions would not seem to be the most important consideration compared to impacts on fish and wildlife. See below regarding habitat fragmentation and wildlife impacts from inundation of the reservoir behind Balbino Dam:

Benchimol, M., and C. A. Peres. 2015a. Predicting local extinctions of Amazonian vertebrates in forest islands created by a mega dam. *Biological Conservation* 187:61-72.

_____. 2015b. Widespread Forest Vertebrate Extinctions Induced by a Mega Hydroelectric Dam in Lowland Amazonia. *Plos One* 10.

2. Are there other studies in the Amazon focused on siting dams optimally with respect to other objectives or criteria, and if so, how did results differ? Hydropower generation is only one ecosystem service produced from dams. This is important because the message related to elevation (higher elevation projects tend to generate more power) changes depending on which function is

considered—for example, if the main purpose of the dams is to store water for irrigated agriculture, lowland dams would be higher priority.

3. Consider providing some context such as guidance for best practices to avoid GHG emissions from damming. For example, logging and removing biomass prior to inundation. Are there others? How would following these best-management practices affect the results?

4. The study uses power generation per unit flooded area to estimate the area of methane emissions associated with each reservoir because ‘emission intensity varies inversely with power generation per unit flooded area.’

- First, it is not explained why GHG emissions would be inversely related to generation. Is it due to correlations with dam head (slope) or flow (drainage area)?

- Second, since the authors estimated flooded area, would it not be possible to separate the effects of power generation and flooded area? For example, to use GIS outlines of reservoirs and bathymetric data that describe the area of shallow perimeter, rather than using a metric that not only includes the whole reservoir area but also confounds reservoir area with generation?

- Third, statistically, using a ratio as a predictor is also not ideal. For example, when generation is low the predictor will be dominated by generation versus flooded area. Some examination of residuals may be in order.

- In Figure 3. It is not surprising to see an inverse relationship here since MWh is in the denominator on the y-axis and in the numerator of the x-axis. You would get such a relationship just by drawing values at random. The same is true for the Pareto-optimal frontier. The two objectives are related because GHG is calculated as an inverse of power density, and therefore is also a function of generation (capacity). This does not invalidate the results (objectives can be correlated), but it makes me wonder if there might be an analytic solution, $d\{C\frac{1}{x} + y/x\}/dx = 0$.

5. Methods: Line 321 on page 13. The authors should consider breaking this into two separate sections, one about the optimization method that exploits tree structure and another section about the trade-off analysis (using a Pareto-optimal frontier). The latter is not, in and of itself, an optimization method (as indicated by the heading).

6. The authors should clarify whether or not the optimization problem assumes spatial dependence among reservoirs. In the optimization, I would have been interested to learn more about how the tree structure was exploited – does the method assume that there is only one dam within a reach? How is it constrained in its search by the structure? I did not have the impression that there was spatial interdependence among the reservoirs in the problem formulation. There is a great deal of detail on benchmarking but very little explaining the method to a non-specialized audience.

7. Line 226 It is not clear where Equation 1 came from. Was it developed in this research or is a reference needed? To improve readability, can the authors move the units of the left-hand-side to the text below and increase the size of the variables in the ratio? I would also suggest adding a subscript to variables that vary by dam to distinguish them from constants. Similarly, notation could be used to indicate that GWP is a function of the time horizon T or better yet, present the general equation as a function of time horizon.

- The 365 T in numerator and denominator cancel, so there is no dependence on time horizon except through GWP. It is also not clear why 365 is in the numerator – should this apply to GWP inside the parentheses?
- More consistency is needed in using italics for variables (or not).
- The downstream emissions estimation seems crude – the same in lowlands and uplands? And the initial GHG contribution from construction is the same for projects of different sizes?

Results

8. The figures, such as figure 1, are nice. However, the map is too small. The authors might consider developing a figure with cumulative emissions (because we care about the total emissions) versus cumulative generation or capacity, the authors could depict how much generation would have to be sacrificed to meet a constraint on total emissions.

9. Figure 2. The figure is fine. However, when people show a Pareto-optimal frontier for a spatial optimization problem, they usually show maps of selected solutions along the frontier (see Polasky et al. 2008. Where to put things? Spatial land management to sustain biodiversity and economic returns. Biological Conservation for example). Consider whether something like this would add valuable information to the paper.

10. The model described in the legend to Extended Data Figure 2 seems to be a mixed model (country is categorical). How was R² estimated? The equation should be written as an equation rather than an R-formula and units should be provided for variables. Also, were model coefficients etc., reported?

11. I would suggest including another figure presenting the main results from Extended Data Table 1 as maps for the 20-y results by lowland/upland within country.

12. There is no Discussion section, but discussion is included with Results. Should this section be labeled 'Results and Discussion'?

13. Page 5 includes a discussion about the need for countries to coordinate efforts to avoid violating 2030 goals. This is odd because the analysis does not include any consideration of spatial relationships, upstream-downstream or grid-related, among hydropower projects. They are all treated as independent. The authors should be sure to mention this somewhere and discuss the implications for relaxing the assumption of independence.

Minor points:

Abstract

1. The sentence, 'Our results show how Andean dam emission intensities are often comparable to solar and wind, whereas many lowland dams exceed intensities of fossil fuel power plants.' is confusing. Can the authors please clarify whether those comparable to solar and wind are upland projects? Also, 'lowland dams' do not 'exceed intensities' – consider rewording this, and please add numerical ranges.

2. In the last sentence in the Abstract, 'clean energy' is not synonymous with 'low-carbon'.

Introduction

3. Please check of provide references and values for the comparison with fossil fuel carbon emissions. Citation 7, Schlomer et al. is not complete. Line 315, reference 7 does not seem to be to the IPCC report.

4. Starting line 112 'Achieving clean energy goals...'. This section does not seem to support the title. I would remove the first paragraph or put it in a Supplemental Info and start with the second paragraph 'Our multi-objective...'. Remove accuracy claims and other computational details around line 120. Note that there are other examples of hydropower optimizations that use exact methods, such as multi-linear integer programming. See examples by

- O'Hanley JR. Open rivers: Barrier removal planning and the restoration of free-flowing rivers. *Journal of Environmental Management*. 2011;92:3112-20.,

- Kuby MJ, Fagan WF, ReVelle CS, Graf WL. A multiobjective optimization model for dam removal: an example trading off salmon passage with hydropower and water storage in the Willamette basin. *Advances in Water Resources*. 2005;28:845-55.
- Zheng PQ, Hobbs BF, Koonce JF. Optimizing multiple dam removals under multiple objectives: Linking tributary habitat and the Lake Erie ecosystem. *Water Resources Research*. 2009;45;
- McManamay, R., JS Perkin, and HI Jager. 2019. Finding convergence among divergent conservation objectives in prioritizing barrier removal in streams. *Ecosphere* 10(2) <http://dx.doi.org/10.1002/ecs2.2596>.

Methods

5. Line 927, the reference looks odd.

6. Page 13 is one long paragraph that includes multiple topics – the authors should remove some of the detail and organize the text better.

7. Is it important to refer to specific policy targets? In general, using a goal like narrows the paper, which will presumably be outdated by 2030.

Results

8. Line 272 starts a new topic and should be a new paragraph.

Reviewer #2 (Remarks to the Author):

This manuscript describes a novel Pareto optimization to compute portfolios of Amazonian dam development that have the highest hydropower capacity per GHG emission. To do this, the authors use existing, previously published, GHG emissions estimates from tropical and subtropical reservoirs together with information about existing and proposed Amazonian dams. They find that upland reservoirs generally emit much less per MWh than lowland systems. They also report significant reductions in overall per MWh emission if future dam development is planned around their

optimization. The manuscript is generally very well written and easy to read. Overall, I think this is a novel and important contribution to the literature (and one that has important policy/management implications). Still, I have some overarching concerns about framing that I believe should be addressed to provide better context (and to better bound uncertainty).

My biggest concern pertains to transparency/clarity in describing how greenhouse gas emissions were estimated from these reservoirs. It is my understanding that the authors used a fixed emission factor based on the database compiled by Deemer et al. 2016, but it is not clear if the emission factor was based on an arithmetic mean, geometric mean or median (or whether bootstrapping considered the full distribution of the dataset)? This information should be made available for ease of interpretation. Also, did the authors only consider methane emission estimates that included both diffusion and ebullition? This would be best practice.

On a related note, I think it is important for the authors to describe the relatively high uncertainty in GHG emission estimates. The authors point out that various environmental factors such as temperature, productivity, and reservoir age can exert an important control on emission, but these factors are not, to my understanding, included in the model of hydropower emissions. Given our uncertainty (and limited environmental data pertaining to the expected productivity of the planned and existing reservoirs) I think this is completely appropriate, but it still needs to be addressed in the paper. The authors should also consider discussing how patterns in reservoir productivity likely bolster their results. If I understand correctly, the patterns reported here (where reservoirs at higher elevations have lower emissions per MWh) are purely based on reservoir morphometry, but higher elevation systems are also likely less productive (e.g. lower catchment area: surface area ratios, lower temperatures, and less developed watersheds). Finally- water level fluctuations have been linked to increased methane ebullition in some systems (Maeck et al. 2014, Harrison et al. 2017, Beaulieu et al. 2017). Is there reason to think that the higher elevation reservoirs would undergo more dramatic water level fluctuations? The authors should at least discuss this possibility.

Finally, the authors should consider providing some broader framing for their study. While they focused on Amazon dams, their approach clearly has the potential to be applicable in other regions. Do the authors feel that the patterns they observed in the Amazon would hold for other basins? A brief description of other regions of the globe undergoing rapid dam construction (and how the findings here may or may not extend to other regions) would be helpful.

Line By Line Edits:

173-183: Important to point out that many dams have other services in addition to hydropower (so a strict per MWh emission may not always be appropriate for weighing costs). Do the higher per MWh emission reservoirs in your study tend to have different secondary purposes than the lower per MWh emission reservoirs?

191-193: I see no citations for the technical and government documents here nor in the supplementary excel file. Please include a list of these documents somewhere.

Signed,

Bridget Deemer

Reviewers' comments:

Reviewer #1:

Review of 'Reducing greenhouse gas emissions of Amazon hydropower with optimal dam planning' by Almeida et al.

The study seeks to limit the GHG emissions of alternative combinations of proposed and existing hydropower project in the Amazon basin to meet a specified target. GHG emissions are estimated for all existing and proposed dams. Estimates are validated and shown to be reasonably good. Reservoirs are shown to vary in GHG emission from less than emissions from wind energy to more than fossil energy sources.

A new multi-objective optimization technique is used to develop a Pareto-optimal frontier of solutions consisting of portfolios of hydropower dams and associated GHG emissions and power capacity. This new optimization method takes advantage of river network structure and has guarantees of producing exact solutions.

Major points

1. The authors should consider reminding readers that there are many other aspects of sustainability that should be considered when planning a configuration of hydropower projects in the Abstract. This is mentioned later, in the 'Moving forward' section. In a hot spot of biodiversity like the Amazon basin, short-term methane emissions would not seem to be the most important consideration compared to impacts on fish and wildlife. See below regarding habitat fragmentation and wildlife impacts from inundation of the reservoir behind Balbino Dam:

Benchimol, M., and C. A. Peres. 2015a. Predicting local extinctions of Amazonian vertebrates in forest islands created by a mega dam. *Biological Conservation* 187:61-72.

_____. 2015b. Widespread Forest Vertebrate Extinctions Induced by a Mega Hydroelectric Dam in Lowland Amazonia. *Plos One* 10.

Response: This is a good point. There are indeed other criteria that need to be considered, as noted in our "Moving forward" section. We have now added this information to the end of the Abstract as well (lines 38-40).

Additionally, we want to stress that we avoid making value judgments on which impacts are the highest priority for decision-makers. Still, we note that GHG assessments are key, especially in early planning stages, because it seems highly undesirable to construct a facility that, in addition to creating a range of social and ecological impacts, does not provide low-carbon energy. We now cite the suggested papers by Benchimol et al. to back up our statement on the other impacts of dams (line 217).

2. Are there other studies in the Amazon focused on siting dams optimally with respect to other objectives or criteria, and if so, how did results differ? Hydropower generation is only one ecosystem service produced from dams. This is important because the message related to elevation (higher elevation projects tend to generate more power) changes depending on which function is considered—for example, if the main purpose of the dams is to store water for irrigated agriculture, lowland dams would be higher priority.

Response: To date, to the best of our knowledge, there has not been an analysis of optimal dam siting at the scale of the entire Amazon basin with respect to any other criteria besides

those described in our work. Even at smaller scales, we could not find multi-objective Pareto optimization works for the Amazon basin in the scientific literature. This approach has been undertaken in other regions like the Mekong (e.g., Schmitt et al. 2018 and Ziv et al. 2012, cited in manuscript). Comparing our results on optimal siting of Amazon hydropower dams with results for other criteria is a very important point and we recognize that in the last paragraph of the main text and the last sentence of the Abstract. Consideration of multiple criteria is a goal of our working group on cumulative impacts of Amazon dams and is a critical next step in our research (<https://impactsofdams.wordpress.com>). In response to the editorial request to frame our work in a wider literature context, which is related to this reviewer's comment, we have added a new paragraph that discusses how strategic planning can lead to better dam decisions (lines 159-169).

We recognize that hydropower generation is only one ecosystem service produced by dams. However, our analysis focuses exclusively on hydropower dams, which is the primary purpose of the projects considered in the Amazon. Thus, our conclusions refer only to dams built for hydropower production, since we analyze the tradeoffs between GHG emissions and electricity production, and the central variable of our analysis is the amount of GHGs emitted per unit electricity produced ($\text{kg CO}_2\text{eq MWh}^{-1}$). We clarify this in the introduction and now mention other dam services in the concluding paragraph (lines 213-214).

3. Consider providing some context such as guidance for best practices to avoid GHG emissions from damming. For example, logging and removing biomass prior to inundation. Are there others? How would following these best-management practices affect the results?

Response: We thank the reviewer for this relevant suggestion. We added a new paragraph that discusses practices that can ameliorate carbon intensities (lines 191-202).

4. The study uses power generation per unit flooded area to estimate the area of methane emissions associated with each reservoir because 'emission intensity varies inversely with power generation per unit flooded area.'

- First, it is not explained why GHG emissions would be inversely related to generation. Is it due to correlations with dam head (slope) or flow (drainage area)?

Response: We believe that there may have been some misunderstanding regarding this issue. Electricity generation capacity per unit flooded area (power density) is not used to calculate total GHG emission, therefore the total GHG emission of a dam is not inversely correlated to power density. What might have led to the confusion here is that two different terms (emission intensity and emission rate/flux) sound very similar. It is important to distinguish GHG emission flux ($\text{kg CO}_2\text{eq km}^{-2} \text{d}^{-1}$) from emission intensities ($\text{kg CO}_2\text{eq MWh}^{-1}$). GHG emission flux gives the evasion of GHG per unit of surface area over a given period of time, which is needed to calculate total GHG emission ($\text{kg CO}_2\text{eq}$ per unit of time). Emission intensity gives total GHG emission per MWh generated, and makes it possible to compare hydropower GHG emission per MWh with those of other energy sources. We recognize that it may be challenging for readers not specialized in reservoir GHG emissions to navigate these terminologies. Because our manuscript may be of interest to a broad audience that includes those interested in system-scale energy planning and optimization algorithms, we added a small table that summarizes the

GHG metrics used in the manuscript (Table 1). In addition, note that we also renamed the metrics to facilitate their understanding. For instance, we replaced “emission intensity” with “carbon intensity”, a widely used synonym that should cause less confusion. With these changes, we hope that it is clearer now that GHG emission is not inversely related to power density. In addition to adding Table 1, we rewrote the second paragraph of the main text to make all concepts clearer.

- Second, since the authors estimated flooded area, would it not be possible to separate the effects of power generation and flooded area? For example, to use GIS outlines of reservoirs and bathymetric data that describe the area of shallow perimeter, rather than using a metric that not only includes the whole reservoir area but also confounds reservoir area with generation?

Response: Separating power generation and flooded area would be ideal, but unfortunately it is not possible. Most proposed dams in the Amazon have limited information available. Project specifications often provide only power capacity and flooded area. Dam height, the spatial configuration of the reservoir, average reservoir depth and other key information for more robust analyses are not provided in the vast majority of cases, and this paucity of information constrains our options for estimating flooded areas. We note, however, that our metric does not confound reservoir area with generation, but instead uses the ratio between these variables (which, in other words, informs the land use intensity of a hydropower project). Dams that flood more land to generate a given amount of electricity tend to result in larger carbon intensities.

- Third, statistically, using a ratio as a predictor is also not ideal. For example, when generation is low the predictor will be dominated by generation versus flooded area. Some examination of residuals may be in order.

Response: We would like to clarify that we performed a bootstrapping procedure to propagate uncertainty, and we do not use a ratio as a predictor. To avoid misinterpretation, we restructured and rewrote our methods section to clarify how we calculate carbon intensities and use bootstrapping to assess uncertainty in GHG emissions (lines 261-308).

- In Figure 3. It is not surprising to see an inverse relationship here since MWh is in the denominator on the y-axis and in the numerator of the x-axis. You would get such a relationship just by drawing values at random. The same is true for the Pareto-optimal frontier. The two objectives are related because GHG is calculated as an inverse of power density, and therefore is also a function of generation (capacity). This does not invalidate the results (objectives can be correlated), but it makes me wonder if there might be an analytic solution, $d\{CI+x+y/x\}/dx = 0$.

Response: As mentioned above, total GHG emission is not inversely correlated to electricity generation or power density. We would like to clarify that the only inputs for the optimization are the total GHG emission (see newly added Table 1 and Equation 1) and hydropower capacity of each dam, and there are no analytical relationships between the two parameters. We added a new figure with the optimization results (Extended Data Figure 1), which is in agreement with another suggestion of Reviewer #1 in a later comment about including a figure with cumulative emissions. After running the optimization, we compute the carbon intensity of each solution to facilitate comparison with other electricity sources (Figure 2).

5. Methods: Line 321 on page 13. The authors should consider breaking this into two separate sections, one about the optimization method that exploits tree structure and another section about the trade-off analysis (using a Pareto-optimal frontier). The latter is not, in and of itself, an optimization method (as indicated by the heading).

Response: We renamed the heading of that Methods section to reflect the content more accurately (“Tradeoff analysis and computation of the Pareto frontier”).

6. The authors should clarify whether or not the optimization problem assumes spatial dependence among reservoirs. In the optimization, I would have been interested to learn more about how the tree structure was exploited – does the method assume that there is only one dam within a reach? How is it constrained in its search by the structure? I did not have the impression that there was spatial interdependence among the reservoirs in the problem formulation. There is a great deal of detail on benchmarking but very little explaining the method to a non-specialized audience.

Response: There are three parts to consider here, and we address them separately below.

1. Spatial dependencies: We do not assume spatial dependencies among reservoirs for hydropower and GHGs. GHGs and electricity generation capacity, the two criteria optimized in our manuscript, are two continuous variables associated with the dams and their reservoirs, and we assume that they do not depend on the underlying river network tree structure. However, as highlighted in our response to Reviewer 1’s second comment, consideration of other criteria besides GHG emissions is a critical next step in our research. Our algorithm has thus been conceived with the broad goal of allowing for optimizing multiple criteria, some of which are heavily dependent on the river network structure (e.g., fish migrations or sediment retention). When such criteria are considered, we really need to reason about the spatial dependence associated with the river network and therefore we use an abstract tree structure, that is also useful for the dynamic-programming approach.

2. Tree structure: Our approach converts the structure of the river network into a more abstract tree structure. Essentially, a node corresponds to a continuous section of the river, uninterrupted by existing or proposed hydropower dams. An edge represents a proposed dam or an existing dam. This tree structure is then used by the dynamic-programming algorithm for the sequence of the merging and pruning of Pareto-optimal solutions. We added more information on the tree structure to our Methods section (lines 398-424).

3. Dynamic-programming approach: The dynamic-programming approach recursively computes the Pareto optimal partial solutions from leaf nodes up to the root. The key insight is that at a given node u , we only need to keep the Pareto-optimal partial solutions and we can therefore eliminate sub-optimal (dominated) solutions. To increase incremental pruning, we convert the original tree into an equivalent binary tree. Given a binary tree, we first compute Pareto optimal solutions for the two children of a given parent node u , enumerate the partial solutions from the children and consider four different combinations of whether to include each of the edges from the children (an edge represents a dam), computing the objective values for the different extended partial solutions and adding them to the set of overall partial solutions. We then remove all dominated partial-solutions from this set. So, the remaining partial-solutions

are Pareto optimal for the parent node. For dealing with memory issues, the algorithm batches partial solutions and is parallelized. We also added more information on the dynamic-programming approach to our Methods section (lines 398-424).

The code that we provide uses the tree structure for the two criteria considered (hydropower and GHGs). We edited the manuscript with the above excerpts and we also provide an explicit pointer to the computer science conference papers for readers interested in additional details (references 23 and 24). We also explicitly indicate that the code is publicly available for download from an online repository (<http://www.cs.cornell.edu/gomes/downloads/hydro-pareto-tree-dp-2c/gomes-selman-et-al-dp-amazon-e-ghg-naturecommunications-2019.zip>; readme file available at <http://www.cs.cornell.edu/gomes/downloads/hydro-pareto-tree-dp-2c/readme.txt>).

7. Line 226 It is not clear where Equation 1 came from. Was it developed in this research or is a reference needed? To improve readability, can the authors move the units of the left-hand-side to the text below and increase the size of the variables in the ratio? I would also suggest adding a subscript to variables that vary by dam to distinguish them from constants. Similarly, notation could be used to indicate that GWP is a function of the time horizon T or better yet, present the general equation as a function of time horizon.

- The 365 T in numerator and denominator cancel, so there is no dependence on time horizon except through GWP. It is also not clear why 365 is in the numerator – should this apply to GWP inside the parentheses?

- More consistency is needed in using italics for variables (or not).

- The downstream emissions estimation seems crude – the same in lowlands and uplands? And the initial GHG contribution from construction is the same for projects of different sizes?

Response: Equation 1 is a conventional way of calculating emission intensities through which one divides all carbon emitted by all energy generated over a given time period. We have made changes to the general formula, including some parameters to provide net emissions, downstream emissions, and construction-related emissions. We now broke the equation into more equations to be more consistent and improve readability. We note that these changes have not modified any results.

We recognize the challenges of downstream emission estimates, which still represent a large knowledge gap in the study of GHG emissions from reservoirs. As highlighted by Prairie et al. 2017 (cited in manuscript) in their review on GHG emission from freshwater reservoirs, downstream emissions are “difficult to predict because they depend on the exact level of the water intake, the vertical CH₄ concentration profile, heterotrophic respiration, bathymetric shape and diffusion-limited oxygen supply to the oxycline”, and “this is a topic for which much more research is required to adequately assess its potential importance and how it varies depending on the particular environmental setting and construction configuration of each reservoir.”. Using different multipliers for lowland versus upland dams would make lowland dams look worse and upland dams look better, which is in agreement with our conclusions; however, we do not have any empirical basis for assuming different multipliers, so we decided to assume a more conservative estimate for upland dams.

Finally, for the GHG contribution from construction, it is important to note that this variable is scaled by electricity generation. Hence, although the emission per MWh is the same for all projects ($19 \text{ kg CO}_2\text{eq MWh}^{-1}$ for a 100-year horizon), dams with larger installed capacities consequently get higher total GHG emissions (i.e., tons of CO_2eq over 100 years, not scaled by MWh).

Results

8. The figures, such as figure 1, are nice. However, the map is too small. The authors might consider developing a figure with cumulative emissions (because we care about the total emissions) versus cumulative generation or capacity, the authors could depict how much generation would have to be sacrificed to meet a constraint on total emissions.

Response: We limited the number of figures in our manuscript to four, as suggested in the guidelines for submission. We included the map as an inset in Figure 1 to comply with the guidelines, and also because we understand that the map helps readers understand Figure 1-a. We have restructured Figure 1 with a larger map that is easier for readers to see important details. In addition, in response to the reviewer's suggestion, we generated a new figure depicting the cumulative emissions versus generation capacity (Extended Data Figure 1). Because we focus primarily on carbon intensities and not total GHG emission in the main text, and also noting the aforementioned space limitations, we opted to include this new figure in the supplementary material.

9. Figure 2. The figure is fine. However, when people show a Pareto-optimal frontier for a spatial optimization problem, they usually show maps of selected solutions along the frontier (see Polasky et al. 2008. Where to put things? Spatial land management to sustain biodiversity and economic returns. Biological Conservation for example). Consider whether something like this would add valuable information to the paper.

Response: This is a great suggestion and we added two new panels in Figure 2 indicating two example solutions, one on the optimal frontier and one that is sub-optimal for the same electricity generation capacity (25 GW).

10. The model described in the legend to Extended Data Figure 2 seems to be a mixed model (country is categorical). How was R^2 estimated? The equation should be written as an equation rather than an R-formula and units should be provided for variables. Also, were model coefficients etc., reported?

Response: The reviewer correctly points out that the regression model presented in the extended figure includes both categorical (country) and continuous covariates (capacity, watershed area, elevation); however, we do not include any random effect variance terms. Thus, our model is a normal multiple linear regression and R^2 is calculated as the coefficient of determination per conventional normal linear regression (e.g. Greene WH. 2012. Econometric analysis. Pearson, New York). To clarify the regression methods in the caption, we provide a more detailed description of the regression model, include units, and specify the form of the model as normal multiple linear regression. We chose a text-based description of the regression model in lieu of providing an explicit formula as the reviewer suggested because notation with categorical variables becomes unwieldy. We included the explicit formula in the Source Data file.

11. I would suggest including another figure presenting the main results from Extended Data Table 1 as maps for the 20-y results by lowland/upland within country.

Response: We thank the reviewer for this suggestion. However, after careful consideration, we concluded that Extended Data Table 1 is the most straightforward way for readers to extract some key country-level information they may be interested in.

12. There is no Discussion section, but discussion is included with Results. Should this section be labeled 'Results and Discussion'?

Response: Thanks for this suggestion. We now labeled the section as Results and Discussion, as suggested.

13. Page 5 includes a discussion about the need for countries to coordinate efforts to avoid violating 2030 goals. This is odd because the analysis does not include any consideration of spatial relationships, upstream-downstream or grid-related, among hydropower projects. They are all treated as independent. The authors should be sure to mention this somewhere and discuss the implications for relaxing the assumption of independence.

Response: The objective of the referred discussion is to demonstrate that achieving sustainable energy relies critically on prioritizing low-GHG dams and avoiding GHG-intensive ones. The idea of using the IEA sustainable development scenario for the global electricity sector in 2040, which is consistent with meeting the 2030 Agenda goals, is to provide readers with a carbon intensity value that is characteristic of sustainable energy. We do not mean to use the 80 kg CO₂eq MWh⁻¹ value as a target that should be met by countries. We reworded the manuscript throughout to clarify that this value is used just as a reference for sustainable energy, and not as a target for countries.

Minor points:

Abstract

1. The sentence, 'Our results show how Andean dam emission intensities are often comparable to solar and wind, whereas many lowland dams exceed intensities of fossil fuel power plants.' is confusing. Can the authors please clarify whether those comparable to solar and wind are upland projects? Also, 'lowland dams' do not 'exceed intensities' – consider rewording this, and please add numerical ranges.

Response: We edited the Abstract to incorporate this helpful reviewer suggestion.

2. In the last sentence in the Abstract, 'clean energy' is not synonymous with 'low-carbon'.

Response: We revised the manuscript and have now avoided the use of "clean energy" throughout.

Introduction

3. Please check of provide references and values for the comparison with fossil fuel carbon emissions. Citation 7, Schlomer et al. is not complete. Line 315, reference 7 does not seem to be to the IPCC report.

Response: The Schlomer et al. citation is indeed an IPCC report (see https://www.ipcc.ch/site/assets/uploads/2018/02/ipcc_wg3_ar5_annex-iii.pdf). We edited the reference list to clarify the citation.

4. Starting line 112 'Achieving clean energy goals...'. This section does not seem to support the title. I would remove the first paragraph or put it in a Supplemental Info and start with the second paragraph 'Our multi-objective...'. Remove accuracy claims and other computational details around line 120. Note that there are other examples of hydropower optimizations that use exact methods, such as multi-linear integer programming. See examples by

- O'Hanley JR. Open rivers: Barrier removal planning and the restoration of free-flowing rivers. *Journal of Environmental Management*. 2011;92:3112-20.,
- Kuby MJ, Fagan WF, ReVelle CS, Graf WL. A multiobjective optimization model for dam removal: an example trading off salmon passage with hydropower and water storage in the Willamette basin. *Advances in Water Resources*. 2005;28:845-55.
- Zheng PQ, Hobbs BF, Koonce JF. Optimizing multiple dam removals under multiple objectives: Linking tributary habitat and the Lake Erie ecosystem. *Water Resources Research*. 2009;45;
- McManamay, R., JS Perkin, and HI Jager. 2019. Finding convergence among divergent conservation objectives in prioritizing barrier removal in streams. *Ecosphere* 10(2) <http://dx.doi.org/10.1002/ecs2.2596>.

Response: We believe that the first paragraph of this section is key to provide some context to readers. As suggested by the reviewer, we removed the comment on line 120 from the main text ("We note that previous heuristic-based hydropower Pareto algorithms do not provide any optimality guarantees") since it may generate confusion. Nevertheless, we would like to clarify our comment concerning the exact computation of the Pareto frontier so that the editors and reviewers understand the rationale for that statement.

The Pareto frontier corresponds to a set of points (X,Y): X represents the value of energy and Y represents the value for GHGs. There are two notions concerning computing the exact Pareto frontier:

- 1) Exact identification of a given point on the Pareto frontier, i.e., for a given value of X an algorithm can find the optimal Y value. Indeed, we fully agree with the reviewer that there are other methods that can compute optimal (exact) points on the Pareto frontier, such as mixed integer programming (MIP). Several researchers, including from our group, have proposed MIP approaches for barrier removal and related problems. We added the references suggested by the reviewer to a newly added paragraph in which we frame our work on a wider literature context (lines 159-169).
- 2) Exact identification of ALL points on the Pareto frontier: Identifying all points on the Pareto frontier cannot be done trivially by a MIP approach, unless one runs a very large number of MIPs. For example, the Pareto frontier represented in Figure 2-a contains 238,459 optimal Pareto points. This is an exact number and the algorithm proves that no other point should be added to the Pareto frontier, i.e., all other solutions are dominated by those 238,459 Pareto frontier points. A MIP approach cannot produce the exact Pareto frontier in the sense of proving that it generates all the Pareto frontier points and no other point belongs to the Pareto frontier. This is essentially because MIP computes one point at a time. Therefore, one would need to run it 238,459 times to find those points, but this would not prove that it finds all the possible points. Our method is quite different since it uses dynamic programming to systematically exploit the

structure of the river network and it (implicitly) considers all possible solutions, eliminating dominated ones. In fact, for a reference, our algorithm for the scenario of Figure 2-a evaluated around 4 billion potential Pareto frontier points, resulting in those 238,459 optimal Pareto frontier points. The algorithm shows that the remaining points are dominated, non-optimal solutions, which therefore do not belong to the optimal Pareto frontier. In addition, the algorithm also proves that no other points need to be evaluated. The algorithm took 20 minutes using 36 threads to run the exact, provably optimal Pareto frontier for our 351 dams.

Methods

5. Line 927, the reference looks odd.

Response: We are unclear about which reference the reviewer refers to here on line 927, since the manuscript ended on line 642.

6. Page 13 is one long paragraph that includes multiple topics – the authors should remove some of the detail and organize the text better.

Response: We broke the paragraph into two and removed some detail, as suggested.

7. Is it important to refer to specific policy targets? In general, using a goal like narrows the paper, which will presumably be outdated by 2030.

Response: As mentioned in the response to comment 13, we revised this throughout. It is not actually a specific policy target, but a reference for sustainable energy.

Results

8. Line 272 starts a new topic and should be a new paragraph.

Response: We agree and have now used a new paragraph.

Reviewer #2:

This manuscript describes a novel Pareto optimization to compute portfolios of Amazonian dam development that have the highest hydropower capacity per GHG emission. To do this, the authors use existing, previously published, GHG emissions estimates from tropical and subtropical reservoirs together with information about existing and proposed Amazonian dams. They find that upland reservoirs generally emit much less per MWh than lowland systems. They also report significant reductions in overall per MWh emission if future dam development is planned around their optimization. The manuscript is generally very well written and easy to read. Overall, I think this is a novel and important contribution to the literature (and one that has important policy/management implications). Still, I have some overarching concerns about framing that I believe should be addressed to provide better context (and to better bound uncertainty).

My biggest concern pertains to transparency/clarity in describing how greenhouse gas emissions were estimated from these reservoirs. It is my understanding that the authors used a fixed emission factor based on the database compiled by Deemer et al. 2016, but it is not clear if the emission factor was based on an arithmetic mean, geometric mean or median (or whether bootstrapping considered the full distribution of the dataset)? This information should be made available for ease of interpretation. Also, did the authors only

consider methane emission estimates that included both diffusion and ebullition? This would be best practice.

Response: We appreciate these comments. To clarify, we have not used a fixed emission factor. We used a bootstrapping procedure that considered the full distribution of the dataset for tropical and subtropical dams for which both diffusion and ebullition data were available (now clarified on lines 300-301). Our bootstrapping procedure randomly sampled 10,000 times from the CO₂ and CH₄ fluxes in the Deemer et al dataset. The uncertainty treatment was previously noted in the last paragraph of the “GHG emission intensity estimates” in Methods, which we have now moved to the beginning of the section (also, this part was slightly reworded to improve clarity). In summary, we used the 10,000 bootstrapped GHG fluxes, whose frequency distribution is based on real-world observations, to estimate ranges of carbon intensities using our Equation 3 given project-specific flooded area (Equation 1) and power generation capacity (Equation 2). This gives us, for a given dam, an expected range of carbon intensities (kg CO₂eq MWh⁻¹) under best-case and worst-case scenarios for GHG emissions (kg CO₂eq km⁻² d⁻¹) (see Extended Data Figure 6 for an example). As noted on lines 307-308, we use the mean values in the main text, and the lower and upper bound of the 95% confidence interval can be found in the Supplementary Material.

On a related note, I think it is important for the authors to describe the relatively high uncertainty in GHG emission estimates. The authors point out that various environmental factors such as temperature, productivity, and reservoir age can exert an important control on emission, but these factors are not, to my understanding, included in the model of hydropower emissions. Given our uncertainty (and limited environmental data pertaining to the expected productivity of the planned and existing reservoirs) I think this is completely appropriate, but it still needs to be addressed in the paper. The authors should also consider discussing how patterns in reservoir productivity likely bolster their results. If I understand correctly, the patterns reported here (where reservoirs at higher elevations have lower emissions per MWh) are purely based on reservoir morphometry, but higher elevation systems are also likely less productive (e.g. lower catchment area: surface area ratios, lower temperatures, and less developed watersheds). Finally- water level fluctuations have been linked to increased methane ebullition in some systems (Maeck et al. 2014, Harrison et al. 2017, Beaulieu et al. 2017). Is there reason to think that the higher elevation reservoirs would undergo more dramatic water level fluctuations? The authors should at least discuss this possibility.

Response: As noted on lines 296-308, the uncertainty in GHG fluxes is captured by our bootstrapping procedure. Our model uses published GHG fluxes to predict a range of 10,000 fluxes that can potentially occur in a given dam, based on real-world observations of the frequency distribution of these fluxes (Extended Data Figure 6). Assuming that we do not know much about the actual GHG fluxes at these two dams in Extended Data Figure 6, if in reality they have low GHG fluxes (e.g., they are particularly oligotrophic), then the observed carbon intensity should be somewhat to the left of the mean. Conversely, if these two dams are particularly eutrophic, the GHG fluxes are expected to be high and the observed carbon intensity should be somewhat to the right of the mean. By running a validation with a small set of Amazon dams, we verify that our model predicts expected carbon intensities well (Extended Data Figure 7).

We know very little about specific project designs, but the fact is that many upland dams tend to be run-of-river design. Thus, we would prefer not to make any speculation as to what to expect for a water level fluctuation versus methane ebullition.

We added a sentence to clarify that nutrient enrichment can lead to higher GHG emission rates, and consequently higher carbon intensities (lines 117-119). We thank the reviewer for pointing this out. Please note that we also added a paragraph on best practices to reduce emissions in response to a relevant comment made by the other reviewer, and reinforced that tackling nutrient inputs that cause reservoir eutrophication can decrease the carbon intensity of a hydropower facility (lines 193-196).

Finally, the authors should consider providing some broader framing for their study. While they focused on Amazon dams, their approach clearly has the potential to be applicable in other regions. Do the authors feel that the patterns they observed in the Amazon would hold for other basins? A brief description of other regions of the globe undergoing rapid dam construction (and how the findings here may or may not extend to other regions) would be helpful.

Response: In response to this helpful comment, we added a broader framing to our results in the concluding section (“Moving forward”).

Line By Line Edits:

173-183: Important to point out that many dams have other services in addition to hydropower (so a strict per MWh emission may not always be appropriate for weighing costs). Do the higher per MWh emission reservoirs in your study tend to have different secondary purposes than the lower per MWh emission reservoirs?

Response: We pointed out that many dams provide other services in addition to hydropower as suggested by the reviewer (lines 213-214). Unfortunately, we do not have information to analyze whether carbon intensity varies differently among dams with distinct secondary purposes. We now note in the Introduction that electricity production is the primary motivation for new dam construction in the Amazon region.

191-193: I see no citations for the technical and government documents here nor in the supplementary excel file. Please include a list of these documents somewhere.

Response: We updated the methods with references to the reports used to update previously published databases utilized to build our database (lines 223-227).

Reviewers' comments:

Reviewer #1 (Remarks to the Author):

The authors have addressed most of my comments adequately, especially providing additional clarification of the methods. I have a few remaining concerns that should be addressed.

1. The maps showing a non-dominant and a dominant solution on the Pareto-optimal frontier are a good addition, but what I was looking for was a characterization and mapping of optimal configurations at different ends of the spectrum. There are many non-dominant solutions and there is no reason to think that the configuration presented has generalizable qualities of interest. One of the new maps shows one end of the spectrum with low installed capacity and low carbon intensity. How does this example differ from maps at the other end of the spectrum with high values of both? An intermediate or 'balanced' solution? These results (specifically the progression from solutions dominated by headwater versus mainstem dams along the frontier) should be discussed as this spectrum is a main result of the paper, including in the Abstract. This is a key result of the paper and I would not recommend publication without it. If the authors have a reason not to want to present this result prominently, then it may require more caveats in the Discussion to explain why.

2. Line 317: The claim here is that practically all methane in the Amazon basin is new and therefore anthropogenic. Please reconcile this claim with the results of the findings below or remove the claim: <https://e360.yale.edu/features/scientists-probe-the-surprising-role-of-trees-in-methane-emissions>

3. Line 164. What is 'sand load'? Do you mean 'sediment loading'? Also, please explain that sediment transport is important to downstream aquatic biota.

4. Line 269: I appreciate that the authors have added dam as a subscript, but suggest using a symbol other than 'd' because 'd' is already being used to denote 'day' in the vicinity of the equation. 2) Also, if there is an equation number, then the equation should be referenced in the text. Alternatively, make them in-line equations. 3) In equation 2, it is not necessary to include the 24 and the units given for Cap_d are wrong (MWh-1, not MW) - unless the symbol Cap_d has been used elsewhere, just give the units as MWd-1 and remove the 24.

5. Please let readers know that, although your optimization algorithm has the ability to solve problems with spatially dependence, this problem did not require it. Perhaps around Line 220 of the 'Moving Forward' section - where you might indicate future need for this ability, or around Line 448 of the 'Methods'.

With these changes, I recommend acceptance.

Reviewer #2 (Remarks to the Author):

The authors have done a nice job responding to the initial rounds of review and I have no further comments or concerns other than a few very minor editorial suggestions:

Line 107: By "larger" you really mean larger surface area right? Not volume? Should probably clarify this.

Line 299: resampled based on a uniform distribution?

Figure 2- It would be nice to visually differentiate the proposed dams from the existing dams in

this figure (rather than just specifying it in the legend)

-Bridget Deemer

Reviewers' comments:

Reviewer #1:

The authors have addressed most of my comments adequately, especially providing additional clarification of the methods. I have a few remaining concerns that should be addressed.

1. The maps showing a non-dominant and a dominant solution on the Pareto-optimal frontier are a good addition, but what I was looking for was a characterization and mapping of optimal configurations at different ends of the spectrum. There are many non-dominant solutions and there is no reason to think that the configuration presented has generalizable qualities of interest. One of the new maps shows one end of the spectrum with low installed capacity and low carbon intensity. How does this example differ from maps at the other end of the spectrum with high values of both? An intermediate or 'balanced' solution? These results (specifically the progression from solutions dominated by headwater versus mainstem dams along the frontier) should be discussed as this spectrum is a main result of the paper, including in the Abstract. This is a key result of the paper and I would not recommend publication without it. If the authors have a reason not to want to present this result prominently, then it may require more caveats in the Discussion to explain why.

Response: Providing a characterization of the solutions along the Pareto frontier is a very nice suggestion. We have generated a new figure that addresses the points raised by the reviewer (new Figure 3). This new figure illustrates six example optimal portfolios along the Pareto frontier (15, 30, 45, 60, 75 and 90 GW). The figure indicates that as total hydroelectric generation increases, optimal portfolios tend to include more dams in lower elevations and larger streams, which are associated with greater carbon intensity (lines 156-159). Some detailed information was also presented in the figure caption to orient readers (lines 660-665). We agree that these findings are key to the paper and have thus emphasized them in the Abstract as well, as suggested (lines 39-40).

2. Line 317: The claim here is that practically all methane in the Amazon basin is new and therefore anthropogenic. Please reconcile this claim with the results of the findings below or remove the claim: <https://e360.yale.edu/features/scientists-probe-the-surprising-role-of-trees-in-methane-emissions>

Response: Please note that the statement mentioned here is referring to reservoirs, and not just for those in the Amazon, but for global reservoirs as discussed by Prairie et al. We now clarify that (line 326) and we added a sentence about the 10% correction that acknowledges the variability in new vs. preexisting emissions (lines 331-334).

3. Line 164. What is 'sand load'? Do you mean 'sediment loading'? Also, please explain that sediment transport is important to downstream aquatic biota.

Response: Sand load refers to sand-sized sediment that is transported by the river (64 μm to 2 mm). Sediment load would not be a precise way of characterizing the fraction of sediments that Schmitt et al. evaluated, because they did not consider suspended sediments smaller than 64 μm and coarse sediments larger than 2 mm. We now clarify what sediment fraction sand refers to, and we note that sand transport is critical for downstream ecosystems (line 166-167).

4. Line 269: I appreciate that the authors have added dam as a subscript, but suggest using a symbol other than 'd' because 'd' is already being used to denote 'day' in the vicinity of the equation. 2) Also, if there is an equation number, then the equation should be referenced in the text. Alternatively, make them in-line equations. 3) In equation 2, it is not necessary to include the 24 and the units given for Cap_d are wrong (MWh⁻¹, not MW) - unless the symbol Cap_d has been used elsewhere, just give the units as MWd⁻¹ and remove the 24.

Response: We thank the reviewer for noting that using *d* as a subscript for dam may lead to confusion because *d* is already denoting day. We have now changed the subscript for dam. As for referencing the equations in the text, note that we already do that when appropriate (e.g, lines 293, 305, and 330, and also in the ExtDataFig_7 spreadsheet provided in the Source Data file).

Regarding the hydropower units, we note that there is a distinction between capacity and energy generation. Installed capacity is a measure of power, not energy, and the unit thus needs to be MW. Conversely, electricity generation is energy production over a time period, which is given in MWh by convention. For example, a 1-MW capacity facility operating at full capacity for an hour will produce 1 MWh of electricity, and 24 MWh d⁻¹ (e.g., <https://www.eia.gov/tools/faqs/faq.php?id=101&t=3>). In Equation 3, because our numerator unit (total GHG flux) is in per day, we need energy generation to be in per day too, therefore we must multiply capacity by 24h in Equation 2. We edited the explanation of Equation 2 to make things clearer (lines 291-297).

5. Please let readers know that, although your optimization algorithm has the ability to solve problems with spatially dependence, this problem did not require it. Perhaps around Line 220 of the 'Moving Forward' section - where you might indicate future need for this ability, or around Line 448 of the 'Methods'.

Response: We thank the reviewer for this suggestion. In order to avoid sacrificing the flow of our "Moving forward" section, we opted to add the suggested edit to the Methods (line 439-443).

With these changes, I recommend acceptance.

Response: We appreciate the comments provided by Reviewer #1 in both rounds of the review process, which we believe have contributed to a stronger manuscript. We hope that the reviewers and the editor will now find our manuscript suitable for publication.

Reviewer #2:

The authors have done a nice job responding to the initial rounds of review and I have no further comments or concerns other than a few very minor editorial suggestions:

We appreciate the comments provided by Reviewer #2 along the review process, which we believe have contributed to a stronger manuscript.

Line 107: By “larger” you really mean larger surface area right? Not volume? Should probably clarify this.

That is right. We thank the reviewer for catching this and changed the manuscript accordingly (line 111).

Line 299: resampled based on a uniform distribution?

Yes. We clarified the manuscript accordingly (line 307).

Figure 2- It would be nice to visually differentiate the proposed dams from the existing dams in this figure (rather than just specifying it in the legend)

Please note that there are no existing dams being shown in any of the panels of our Figure 2.

Figures 2-a,b,c,d show only portfolios of dams, whereas Figures 2-e,f show proposed dams based on two example portfolios extracted from panel b.

REVIEWERS' COMMENTS:

Reviewer #1 (Remarks to the Author):

The revised manuscript appears suitable for publication. I am happy to be identified as Dr. Henriette Jager.